# Liner-Shipping Network Design with Emission Control Areas: A Real Case Study

**Xiangang Lan** [1,*], **Qin Tao** [2] **and Xincheng Wu** [3]

1    Zhuhai City Polytechnic, Zhuhai 519090, China
2    Faculty of Business, City University of Macau, Macau 999078, China
3    Navigation College, Jimei University, Xiamen 361021, China
*    Correspondence: lxg1988@stu2017.jnu.edu.cn

**Abstract:** In recent years, liner-shipping companies have faced a traditional trade-off between cost and emission ($CO_2$ and $SO_X$) reduction. This study considers this element to construct a liner-shipping network design model which includes a package-cargo transport plan, route allocation, and route design. The objective is to maximize profit by selecting the ports to be visited, the sequence of port visits, the cargo flows between ports, and the number/operating speeds of vessels. In addition, emission control areas (ECAs) exist in the liner network. With reference to the idea of the column generation algorithm, this study proposed a heuristic algorithm based on empirical data through a real case calculation and selected the optimal scheme, which is in-line with both economic and environmental benefits. The results show that the model and optimization method are feasible and provide an effective solution for the liner network design of shipping companies, while also considering environmental factors. In addition, the effects of the number of ECAs, inter-port origin-destination (OD) demand, freight rate, fuel price, and carbon prices on the design of transport networks are discussed to provide a reference for the operation of shipping companies and government decision-making.

**Keywords:** liner-shipping network design; emission control areas (ECAs); real case; heuristic algorithm based on empirical data; shipping companies

## 1. Introduction

Global warming is a major environmental problem faced by humankind. According to the Intergovernmental Panel on Climate Change (IPCC), an intergovernmental body of the United Nations, if the global greenhouse effect continues at the current rate, the global average temperature by 2030 will be more than 1.5 degrees higher than it was during the pre-industrial era [1]. In August 2020, the International Maritime Organization (IMO) released its fourth greenhouse-gas study [2]. The report stated that between 2012 and 2018, $CO_2$ emissions from the world's shipping industry increased from 977 million tons to 1076 million tons, accounting for approximately 3.1% of total global $CO_2$ emissions. In addition to $CO_2$ emissions, other exhaust pollution problems caused by ships should not be underestimated. PM2.5, $SO_X$, $NO_X$, and other harmful pollutants contained in ship exhaust can pose a significant threat to human health and the environment. The impact of the ship transport industry on the environment is substantial. Liner-shipping network optimization which considers environmental factors has attracted the attention of many countries, enterprises, and academics. This study highlights four major sulfur emission control areas (SECA) which have been established in European and American waters since 2012. In China, the emission control areas (ECAs) were clearly defined on 9 July 2018, in the official announcement of the ECAs program. Awareness of environmental protection has increased in all countries. What challenges does the implementation of ECAs bring to liner network optimization? Windeck [3] described the design of a liner-shipping network,

considering the ports of call, transported cargo, and number of available vessels, while also considering the environmental impact. Such studies have drawn some interesting conclusions, which provide a theoretical basis for this study.

However, few existing studies focus on the optimization of container-liner transportation with a comprehensive consideration of environmental factors, including ECAs. In fact, organizations such as the International Maritime Organization (IMO), North America, and the European Union have set up emission control zones to protect humanity from air pollution from ships. The navigation operations and other activities of the ships are strictly controlled in the control area. Ships are required to use clean fuel when sailing, anchoring, and docking in the control area. The price of clean fuel is bound to be much higher than that of ordinary fuel, which leads to higher fuel costs and affects transportation optimization. How can economic, fleet transport efficiency, and environmental benefits be balanced in the context of ECAs to accommodate stricter future environmental policies? These issues warrant further research. To bridge these gaps, we address the following questions.

- The docking port, docking order, ship type, quantity, and speed are the decision variables. Considering environmental factors, how can a liner-transportation network optimization model be built?
- Liner-transportation network optimization is an NP-hard problem. What algorithm should be designed to solve the liner-transportation network optimization considering environmental factors?
- What is the impact of the ECA policy on shipping-company profits and transportation networks?

To answer these questions, the environmental impact was transformed into the environmental cost of marine transportation and incorporated into its cost structure. The model was transformed into a two-stage combined-speed optimization, and the liner-transportation network design problem was constructed as an upper and lower level planning model. The route-design problem (docking port and port calling sequence) was set as the upper level design problem, and the route-operation optimization (including route allocation and the route transportation plan) was set as the lower level problem. Referring to the idea of a column generation algorithm, a heuristic algorithm based on empirical data was proposed, and the model was decomposed into a route-generation model and a route network-optimization model, including four different cases to calculate the example. It also analyzes the influence of fluctuations in port OD demand, freight rate, ECAs, and other factors on the route network and its profit. At the same time, it compares the adjustment scheme of the route network and the change in total profit when using the above methods.

This study has several theoretical and practical implications. It proposes a linear network optimization model considering environmental costs, designs a heuristic algorithm based on empirical data, and proves the effectiveness of the model and algorithm through empirical calculation, which provides an effective solution for the liner network design of shipping companies. The heuristic algorithm based on empirical data proposed in this study considers the problem from the perspective of global optimization and can obtain a better global solution than other methods. Moreover, this study offers the following valuable insights:

- The implementation of the ECA policy had little impact on the profits of shipping companies and transportation networks. Although the establishment of an emission control zone will increase the cost burden of shipowners to a certain extent, the impact of ECAs on the cost is relatively small compared to the total.
- The ECA policy has little influence on port attractiveness and competitiveness, and port cargo supply and demand, namely, port throughput, is the key factor in attracting ships.
- Freight rate fluctuation has the most significant impact on the shipping company's profit and transportation network, followed by the impact of inter-port OD demand

fluctuation. Increasing port cargo volume is key to increasing port attractiveness and competitiveness.

The remainder of this paper is organized as follows. Section 2 presents a literature review. Section 3 introduces model specifications, notations, and key assumptions. The models are presented in Section 4. The proposed algorithm is outlined in Section 5, and a case study and summary conclusions are presented in Sections 6 and 7, respectively.

## 2. Literature Review

This study is related to two research streams: liner-transport network optimization and optimization algorithm design for the liner transport network.

### 2.1. Liner Transport Network Optimization

Brouer et al. [4] built a library of standard algorithms involving optimal design models for linear shipping. Du et al. [5] proposed a mixed-integer linear programming model which considers collaborative transportation. The model is efficiently solved by C++ and is called ILOG CPLEX. Cheaitou [6] proposed a multi-objective optimization (MOO) model for liner shipping based on profit maximization and $CO_2$ and $SO_X$ emission minimization, wherein all the objective functions of the model are vessel speed functions. Pierre et al. [7] built a mixed-integer linear programming (MILP) model for a container-liner-shipping network design which considers the trade-off between the minimization of costs and the minimization of $CO_2$ and $SO_X$ emissions. Zhu et al. [8] studied the fleet planning problem under uncertain carbon tax policies and developed a multistage stochastic integer programming model for liner carriers. Branchini et al. [9] coordinated the assignment decisions of ships to contractual and spot voyages and the determination of ship routes and schedules to maximize profit. Gelareh [10] constructed a mixed-integer programming method for designing a hub-and-spoke network in a competitive environment. Gelareh and Pisinger [11] built a mixed-integer linear model considering route network design, hub port selection, and fleet planning and designed a decomposition algorithm to solve it. Meng and Wang [12] established a linear optimization model of a container liner network based on a central network, multi-port docking, and empty container distribution under the assumption of constant speed and used the CPLEX solution package to solve the model. Ameln et al. [13] analyzed a new formulation of the liner-shipping network design problem based on a two-layer network, which considers trans-shipment costs and takes into account the complex service structure, and proposed valid inequalities and a novel approach to inner representations of low-dimensional polyhedra. Xing et al. [14] studied container-ship speed optimization and fleet scheduling problems in the context of two carbon-emission policies: carbon cap-and-trade and carbon tax. Ma et al. [15] defined the cost of carbon emissions; the route network, speed, and refueling strategy were optimized to minimize the total voyage cost. Zhen [16] proposed a bi-objective mixed-integer linear programming model, aiming to optimize sailing routes and speeds within and outside ECAs while minimizing the total fuel cost and $SO_2$ emissions. A new algorithm was developed to solve the proposed model by combining the two-stage iterative algorithm and fuzzy logic method based on the $\in$ constraint. Xin et al. [17] aimed to minimize generalized transportation costs and established a joint optimization model for shipping network design and infrastructure investment. Ma et al. [18] developed a ship routing and speed MOO model which considers ECAs. Gao and Hu [19] established a multi-objective MILP model to optimize the allocation of liner routes with multiple ship types on multiple routes. Zhuge et al. [20] presented a mixed-integer non-linear programming model to minimize the total cost, including the fixed cost of ships deployed, bunker tank cleaning cost, and bunker cost. Wang and Meng [21] constructed a mixed-integer nonlinear programming model to study the optimal speed of container ships on each route in a liner transportation network. Song et al. [22] considered the optimization problems of ship allocation, planned sailing speed, and service adjustment of liner routes with uncertain port times based on expected cost, service reliability, and shipping emissions. Chuang et al. [23] proposed a fuzzy genetic

algorithm for container-liner transportation planning in 2010 to solve the optimization problem of multiport-affiliated ship transportation networks with uncertain demands. These studies provide a solid foundation for our study, but few of them comprehensively consider environmental factors, including ECAs.

### 2.2. Optimization Algorithm Design of Liner Transport Network

Plum et al. [24] proposed a new algorithm for the logistics-service network design problem. Brouer et al. [25] designed a heuristic algorithm based on integer programming for liner-shipping network design. Karsten et al. [26] considered the coordination between ships and constraints on the transit time for cargo movement, proposed a mathematical model for the problem of container-liner-shipping network design, designed an improved heuristic algorithm to solve it, and established an MOO model for liner route allocation and cargo allocation. Cheng and Wang [27] proposed a hybrid genetic algorithm to solve the container-liner-shipping network design problem. Ma et al. [28] designed a route and speed optimization method to simultaneously reduce sailing cost and time, considering the regulations of ECAs and weather conditions. The non-dominated Sorting Genetic Algorithmsorting genetic algorithm was used to determine the Pareto optimal. Wang et al. [29] addressed a holistic liner-shipping service planning problem which integrated fleet deployment, schedule design, sailing path, and speed optimization, considering the effect of ECAs, and proposed a nesting algorithmic framework to address this new and challenging problem. Sheng et al. [30] developed a mixed-integer convex cost-minimization model for the determination of optimal vessel speeds and fleet size and developed an analytical optimal solution for the model by relaxing the integer restriction of the fleet size variable. The above studies have laid a foundation for solving the linear-network transport optimization model, but there is still a gap: solving the linear-network optimization model while taking comprehensive consideration of environmental factors, including carbon-emission control and sulfur-emission control.

The container-liner transportation network optimization design (LSND) includes a course design (affiliated port choice and affiliated order), the ship plan (ship-type selection, fleet-number input, and speed), and shipping goods-transport plan as three sub-problems in comprehensive problem sets; many researchers have considered these three sub-problems, respectively, in the related research. In recent years, with the development needs of the international shipping industry and the continuous deepening of academic research, the research focus of liner-transport network optimization design has gradually developed from decentralized research to focus on the unified correlation and comprehensive consideration of these three sub-problems, to obtain the overall optimal transport network. Zhen et al. performed route and speed optimization considering ECAs. Wang et al. performed a good study on transportation network optimization; however, they all considered the same speed inside and outside the ECA. On this basis, this study considers two-stage speed (different speeds inside and outside the ECA) transport network optimization considering the course design, ship plan, and shipping-goods transport plan.

### 3. Model Specifications and Parameter Definition

The LSND consists of three subproblems: cargo transportation planning, route allocation, and route design. Agarwal et al. were the first to consider these three sub-problems together and proposed various algorithms to solve the LSND problem [31]. In liner shipping, ship carbon emissions depend on ship fuel consumption; therefore, carbon emissions affect not only the optimal speed and the fixed operating cost of the ship but also the design of the route and the choice of cargo. In addition, the sulfur-emission control factor is based on ECAs, which transform the model into a two-speed optimization model. As shown in Figure 1, this study takes the port of call, port-call sequence, type and number of vessels, and internal and external speeds of the ECAs as decision variables and integrates the three interrelated sub-problems of liner-shipping network design with the goal of the overall optimization of the shipping network.

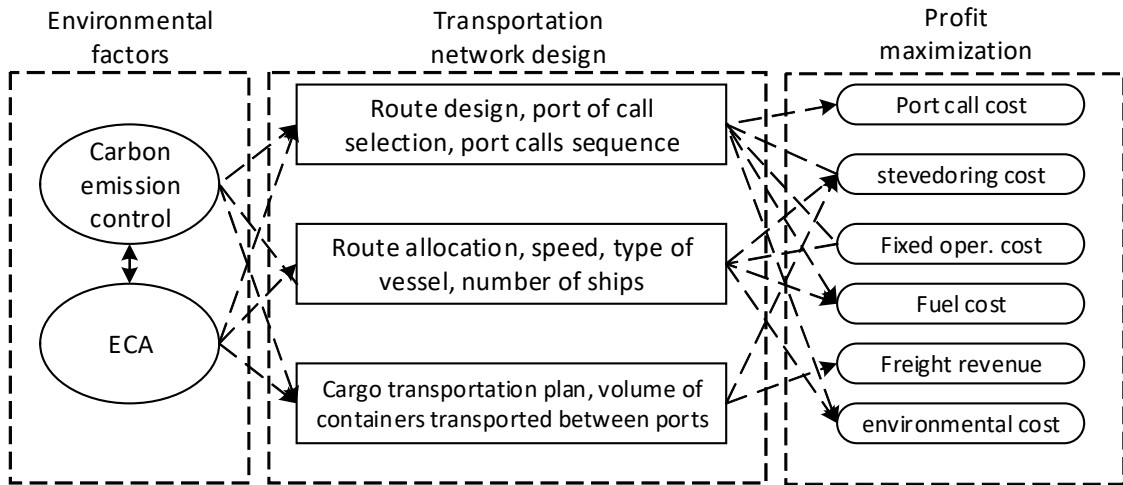

**Figure 1.** Diagram of the research issues.

Assumptions: (1) ships with capacities greater than the freight volume are idle, regardless of the shipping investment; (2) the problem of liner-shipping optimization worsens from the perspective of ship operating efficiency, and it considers that ships navigate normally without any unusual circumstances; that during the voyage, no accidents of any sort occur; that the ship is operating on a simple voyage; and that each route has a weekly frequency, with the same route and speed; (3) there is stable international trade during the research process, so the OD transport demand between ports on the route remains unchanged; (4) the ship size on the route is determined by the OD demand between ports and OD transport demand remains unchanged, so the same ship type is put on the same route but the choice of ship type on the route is made through model optimization; (5) technological development is improving port conditions and this study selected famous ports of the world with developed infrastructures, so it can be assumed that the efficiency of the loading and unloading operations, port charges, and the arrival and departure times of ships of each port in the route are the same; (6) according to the usage of trade, ship loading and OD demand between ports are measured in TEU; (7) for convenient modeling, two virtual ports, 0 and N + 1, were set as the starting and ending ports of the route, as shown in Figure 2.

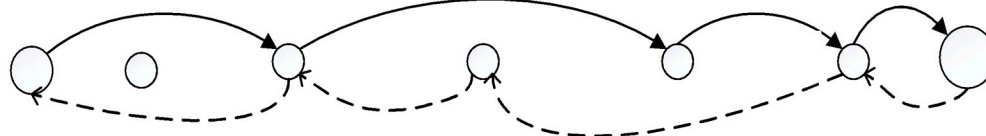

**Figure 2.** Diagram of research issues.

- Description of the variables

  $R$ is the integration of all routes.
  $N = \{1, 2, 3 \dots n\}$ is the integration of the ports in the transport network.
  $N^e = \{0, 1, 2, 3 \dots n, n + 1\}$ is the integration of the ports in the virtual transport network.

- Decision variables

  $x^r$ is a 0, 1 variable whose value is 1 when route $r$ is selected as the optimal route network, and 0 otherwise.
  $y_{ij}^r$ denotes the volume of containers (*TEU*) carried on route $r$ between ports $(i, j)$.
  $v_{rm}^{out}$ denotes the speed of navigation outside the *ECAs* of a ship of type $m$ on route $r$.
  $v_{rm}^{ECA}$ denotes the speed of navigation within the *ECAs* of a ship of type $m$ on route $r$.
  $n_m^r$ is the number of $m$ vessels assigned on route $r$.

$x_m^r$ is a 0, 1 variable whose value is 1 when the route is configured with an $m$ vessel and 0 otherwise.

$z_{ij}^r$ is a 0, 1 variable, and *1* means that the ship on route $r$ goes directly from port $i$ to port $j$. Otherwise, $z_{ij} = 0$.

- Port-related parameters

    $D_i^{ECA}$ is the $i$-port distance from ECAs (*nm*).
    $D_j^{ECA}$ is the $j$-port distance from ECAs (*nm*).
    $D_{ij}$ is the voyage between ports $i$, $j$.
    $q_{ij}$ is the weekly cargo demand (*TEU*) between ports $i$ and $j$.
    $N$ is the number of ports of call on the route.
    $r_{ij}$ is the freight rate for shipments between ports $i$ and $j$ (USD/TEU).
    $t_{pil}$ is the time required for the berthing and unberthing of a ship entering and leaving the port.
    $l$ is port loading and unloading efficiency (TEU/hour);
    $C_m$ is the fixed daily rate for the $m$ vessel type, also called the daily charter rate (USD/day).
    $C^l$ is port charges for loading containerized cargo (USD/TEU).
    $C^u$ is port charges for unloading containerized cargo (USD/TEU).

- Ship-related parameters

    $T_r$ is the total time taken by a single vessel on route $r$ *to* complete a round-trip voyage.
    $F^r$ is the total daily fuel consumption of the fleet on route $r$ (tons).
    $Q_{CO_2}^r$ is the weekly average $CO_2$ emissions of the fleet on route $r$ (tons).
    $C_{CO_2}^r$ is the cost of carbon emissions on route $r$.
    $f^r(tp)$ is the profit of the fleet on route $r$ (USD).
    $F_m$ is the daily fuel consumption of the m-vessel's main engine at design speed (t/day).
    $A_m$ is the daily fuel consumption of the m-vessel's auxiliary engine at design speed (t/day).
    $P_{IFO}$ is the heavy-crude-oil price (USD/ton).
    $P_{VLSFO}$ is the very-low-sulfur fuel oil price (USD/ton).
    $\lambda$ is the carbon conversion factor.

## 4. Models

The liner transportation network design uses a two-layer model for problem framing, as shown in Figure 3, setting the route design problem (port of call and port-of-call sequence) as the upper level, and the route optimization problem (including route allocation and transportation plan) as the lower level. There is no unified standard calculation method for measuring sulfur emission costs. In terms of marine sulfur emissions, there is no clear "marine sulfur tax," and the main control method of sulfur emissions in the shipping market is the ECAs; thus, the cost of sulfur emission was replaced by the increase in fuel cost due to the ECAs. The environmental cost of maritime transport is included in the cost structure of maritime transport.

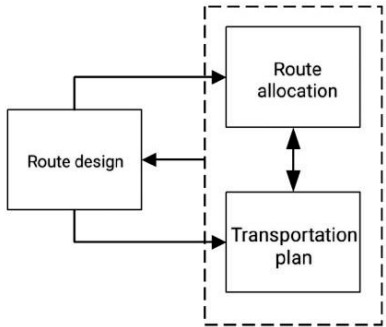

**Figure 3.** Two-layer optimization model.

### 4.1. Route Operating Cost

- Total cargo transport between ports

$Q_{ij}$ is the total cargo volume transported between port *i* and port *j*. For example, as shown in Figure 4, the total volume transported between port 1 and port 4 includes the amount transported from port 1 to port 4 and the amount transported from port 1 to port *n*. As the figure assumes that the solid line represents the direction of export, the total cargo transport between ports in the export direction is $Q_{ij} = \sum_{o=1}^{i} \sum_{p=j}^{n} y_{op}$; the dashed line represents the import direction, and the total cargo transport between ports in the import direction is $Q_{ji} = \sum_{o=1}^{i} \sum_{p=j}^{n} y_{po}$.

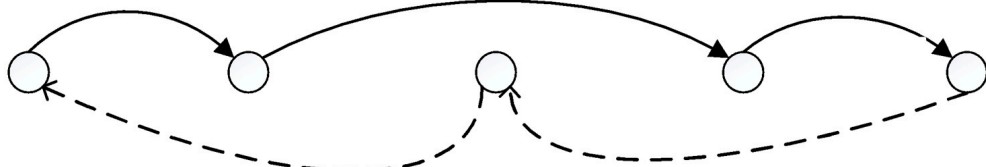

**Figure 4.** Segments of total cargo volume.

- The connection between navigation speed, navigation time, and the number of vessels. Let the sailing time between ports be $t_{ij}^{r}$, equal to the voyage divided by the speed, and the expression is:

$$t_{ij-sailing}^{r} = x_m^r \cdot z_{ij}^r \left( \frac{D_{ij} - D_i^{ECA} - D_j^{ECA}}{v_{rm}^{out}} + \frac{D_i^{ECA} + D_j^{ECA}}{v_{rm}^{ECA}} \right) \tag{1}$$

Let the time in port be $t_{ij}^{r}$, including loading and unloading time and berthing and unberthing time, and the expression is:

$$t_{ij-mooring}^{r} = z_{ij}^r \left( \frac{2y_{ij}^r}{l} + 2t_{pil} \right) \tag{2}$$

The total time for a single ship to complete a round-trip voyage is an expression consisting of the time at sea, the time spent loading and unloading at the port, and the pilotage time while entering and leaving the port, and the expression is:

$$T^r = \sum_{m \in M} \sum_{i \in N} \sum_{j \in N} x_m^r \cdot z_{ij}^r \left( \frac{D_{ij} - D_i^{ECA} - D_j^{ECA}}{v_{rm}^{out}} + \frac{D_i^{ECA} + D_j^{ECA}}{v_{rm}^{ECA}} + \frac{2y_{ij}^r}{l} + 2t_{pil} \right) \tag{3}$$

A week has 7 days (24 h/day), which amounts to 168 h, and the number of ships allocated per week is equal to the total voyage time divided by 168. The expression for the number of ships allocated, *n*, is:

$$n_m^r = \frac{T^r}{168} = \sum_{m \in M} \sum_{i \in N} \sum_{j \in N} \frac{x_m^r \cdot z_{ij}^r}{168} \left( \frac{D_{ij} - D_i^{ECA} - D_j^{ECA}}{v_{rm}^{out}} + \frac{D_i^{ECA} + D_j^{ECA}}{v_{rm}^{ECA}} + \frac{2y_{ij}^r}{l} + 2t_{pil} \right) \tag{4}$$

Let the fleet revenue, excluding stevedoring, be $f(r)^r$ and the expression for the freight rate minus the stevedoring rate multiplied by the volume as the fleet revenue is:

$$f(r)^r = \sum_{i \in N} \sum_{j \in N} \left( r_{ij} - C^l - C^u \right) \cdot y_{ij}^r \tag{5}$$

- Fleet costs

Fleet costs include operating costs, port charges, and fuel consumption costs. Let the weekly operating costs of the fleet be $C_{op}^r$, the average port charges per week be $C_p^r$

(including mooring and unmooring fees, port dues, and other fixed charges), and the average weekly fuel consumption costs be $C_v^r$. Then, the expressions for $C_{op}^r$, $C_p^r$, and $C_v^r$ are:

$$C_{op}^r = \sum_{m \in M} (7 \cdot x_m^r \cdot C_m \cdot n_m^r) \tag{6}$$

$$C_p^r = \sum_{m \in M} \sum_{i \in N} \sum_{j \in N} x_m^r \cdot z_{ij}^r \cdot G_i^m \tag{7}$$

Fuel consumption is proportional to the cube of the speed. $F_m \left( \frac{v_m}{v_m^0} \right)^3$ indicates the fuel consumption of the ship's main engine. The main and auxiliary engines of a ship use different fuels, usually heavy oil for the main engine and light oil for the auxiliary engines.

$$F^r = \sum_{m \in M} \sum_{i \in N} \sum_{j \in N} x_m^r \cdot z_{ij}^r \cdot \left( F_m \left( \frac{v_{rm}^{out}}{v_m^0} \right)^3 \frac{D_{ij} - D_i^{ECA} - D_j^{ECA}}{168 v_{rm}^{out}} + F_m \left( \frac{v_{rm}^{ECA}}{v_m^0} \right)^3 \frac{D_i^{ECA} + D_j^{ECA}}{168 v_{rm}^{ECA}} \right) + A_m \cdot n_m^r \tag{8}$$

$$C_v^r = \sum_{m \in M} \sum_{i \in N} \sum_{j \in N} x_m^r \cdot z_{ij}^r \left( P_{IFO} \cdot F_m \left( \frac{v_{rm}^{out}}{v_m^0} \right)^3 \frac{D_{ij} - D_i^{ECA} - D_j^{ECA}}{168 v_{rm}^{out}} + P_{VLSFO} \cdot F_m \left( \frac{v_{rm}^{ECA}}{v_m^0} \right)^3 \frac{D_i^{ECA} + D_j^{ECA}}{168 v_{rm}^{ECA}} \right) + A_m \cdot n_m^r \cdot P_{VLSFO} \tag{9}$$

- Cost of carbon emissions

The relationship between speed, fuel consumption, and carbon emissions: ship carbon emissions depend on ship fuel consumption in a certain period and the carbon emission factor of fuel $\lambda$. The carbon emission factor varies slightly in the literature. In this paper, we used the factor $\lambda = 3.114$ from the Fourth Greenhouse Gas Study 2020, that is, 1 t of marine fuel produces 3.114 t of $CO_2$. Therefore, the average weekly $CO_2$ emissions of the fleet are $Q_{CO_2}^r$. There are two main potential carbon-emissions policies available: the cap-and-trade program and the carbon tax [14,32–34], according to the carbon tax on shipping emissions, assuming a $\delta$ fixed carbon tax per ton (USD/ton), the cost of carbon emissions $C_{CO_2}^r$ is:

$$C_{co_2}^r = Q_{co_2}^r \cdot \delta = \lambda \cdot F^r \cdot \delta \tag{10}$$

Based on the above assumptions, parameter definitions, and related calculations, the problem was framed with a mathematical model.

$$C_{co_2}^r = Q_{co_2}^r \cdot \delta = \lambda \cdot F^r \cdot \delta \tag{11}$$

### 4.2. Route Design

The liner route selection needs to solve two important problems, namely, the port-of-call selection and the port-of-call sequence optimization. This is a typical KP problem, which is a classic problem of operations research. This paper develops the following liner-route selection model.

$$maxZ_{TP} = \sum_{r \in R} x^r \cdot f(tp)^r \tag{12}$$

### 4.3. Liner Transportation Network Design

Based on the above analysis, an optimization model of the liner-shipping network which considers the environmental costs is expressed as follows:
Objective function:

$$maxZ_{TP} = \sum_{r \in R} x^r \left\{ f(r)^r - C_{op}^r - C_p^r - C_v^r - C_{co_2}^r \right\} \tag{13}$$

Constraints:

1.  Capacity constraints

$$\sum_{r \in R} x^r \cdot y_{ij}^r \leq D_{ij} \tag{14}$$

2.  Minimum one leg of the route

$$\sum_{i=1}^{n-1} \sum_{j=i+1}^{n} \left( z_{ij}^r + z_{ji}^r \right) \geq 1, \forall r \in R \tag{15}$$

3.  Closed-loop route

Export direction:

$$\sum_{i=0}^{p-1} z_{ip} - \sum_{j=p+1}^{n+1} z_{pj} = 0, \forall p \in N; r \in R \tag{16}$$

$$\sum_{j=1}^{n} z_{0j}^r = 1, \forall r \in R \tag{17}$$

$$\sum_{i=1}^{n} z_{i,n+1}^r = 1, \forall r \in R \tag{18}$$

Import direction:

$$\sum_{i=0}^{p-1} z_{pi}^r - \sum_{j=p+1}^{n+1} z_{jp}^r = 0, \forall p \in N; r \in R \tag{19}$$

$$\sum_{j=1}^{n} z_{j0}^r = 1, \forall r \in R \tag{20}$$

$$\sum_{i=1}^{n} z_{n+1,i}^r = 1, \forall r \in R \tag{21}$$

4.  Route allocation constraints

$$\sum_{m \in M} x_m^r = 1, \forall r \in R \tag{22}$$

5.  Cargo transportation and port-of-call constraints

Export direction:

$$y_{ij}^r \leq D_{ij} \sum_{p=i+1}^{j} z_{ip}^r, \forall i, j \in N; i < j; r \in R \tag{23}$$

$$y_{ij}^r \leq D_{ij} \sum_{p=i}^{j-1} z_{pj}^r, \forall i, j \in N; i < j; r \in R \tag{24}$$

Import direction:

$$y_{ij}^r \leq D_{ji} \sum_{p=i}^{j-1} z_{jp}^r, \forall i, j \in N; i < j; r \in R \tag{25}$$

$$y_{ij}^r \leq D_{ji} \sum_{p=i+1}^{j} z_{pi}^r, \forall i, j \in N; i < j; r \in R \tag{26}$$

6. Vessel capacity limit, with W being a very large positive number

$$Q_{ij}^r \leq \sum_{m \in M} B_m x_m^r + W\left(1 - z_{ij}^r\right), \forall r \in R \tag{27}$$

$$Q_{ji}^r \leq \sum_{m \in M} B_m x_m^r + W\left(1 - z_{ji}^r\right), \forall r \in R \tag{28}$$

7. Vessel speed constraint

$$v_m^{\min} \leq v_{rm}^{ECA} \leq v_{rm}^{out} \leq v_m^{\max}, \forall r \in R \tag{29}$$

$$\sum_{r \in R} x^r = 1 \tag{30}$$

With the port of call, sequence of port calls, the type and number of vessels, and the speed inside and outside the ECAs as decision variables, the objective function (6)–(13) indicates that the maximum weekly profit of the fleet consists of five parts. The first part is the freight revenue minus the loading and unloading costs; the second part is the fixed operating cost of the selected vessels; the third part is the fixed cost of the port; the fourth part is the fuel cost (implicitly including the cost of sulfur emissions from the ECAs), and component 5 is the cost of carbon emissions.

The constraints are divided into seven parts.

(14) A volume constraint, i.e., the volume does not exceed the transport demand between ports $(i, j)$ on the route.

(15) Route segments greater than 1.

(16) and (21) Closed-loop route: (16) indicates that in the outbound route, the ship enters and exits a port an equal number of times (0 or 1); (17) and (18) indicate that in the outbound direction, the ship's route starts at virtual port 0 and ends at port $n + 1$; (19)–(21) indicate the same constraints on the import direction.

(22) Route allocation constraint, with the same type of vessel on the route.

(23)–(26) Cargo transport and port-call constraints: if there is cargo transport between ports $(i, j)$, the ship must perform a port call $(i, j)$.

(27)–(28) Vessel capacity limits; i.e., the amount of cargo transported is not allowed to exceed the maximum capacity of the vessel.

(30) Vessel speed constraint; the speed in the ECAs is less than the speed outside the ECAs, and the speed respects the maximum and minimum speed limits of the vessel.

## 5. Arithmetic Design

### 5.1. Overall Design of the Algorithm

The model is a mixed-integer nonlinear programming problem with quadratic and inverse quadratic terms that is difficult to solve directly. Column generation is an effective method for solving large-scale linear programming problems, particularly in large-scale models. The fundamental idea originated from the decomposition strategy proposed by Dantzing and Wolfe in 1960, which is based on the theory of convex programming for a class of linear programming models with angular structures. This is referred to as the Dantzing–Wolfe decomposition. The main idea is that the initial linear programming problem is decomposed by dividing it into a master problem and a subproblem. Then, a restricted master problem is devised by searching for some variables from the master problem (up to the point of obtaining at least one feasible solution). The restricted master problem is then solved until an optimal solution is obtained, the resulting pairs of variable values are used for the subproblem, and the subproblem is solved to form columns with a positive reduced cost (the objective function is the maximum). Then, the column with the maximum positive reduced cost is added to continue the solution. The above process is

repeated until a column with a positive reduced cost cannot be generated, at which point the original problem is optimized.

Referring to the concept of column generation, this study proposes a heuristic algorithm based on empirical data. The model is decomposed into a route generator model and route-network optimization model. The design of the algorithm is shown in Figure 5. For the route generator model, we first calculated the profit of the initial route (a record), given the allocation type m and speed value $V_6^{ECA} = V_6^{ECA}out = V^0$, setting the actual route of current operations of the enterprise as the initial route. We then compared the profit of the next feasible route according to the set parameters. If it was greater than the profit of the initial route, it was included in the set of candidate routes and was repeated until all routes were calculated, recording the ports of call and sequence of port calls. For the route-network optimization model, the routes in the candidate route library were optimized to choose the best route, and the type and number of ships, internal and external speeds of ECAs, and cargo volume between ports configured on the route were calculated.

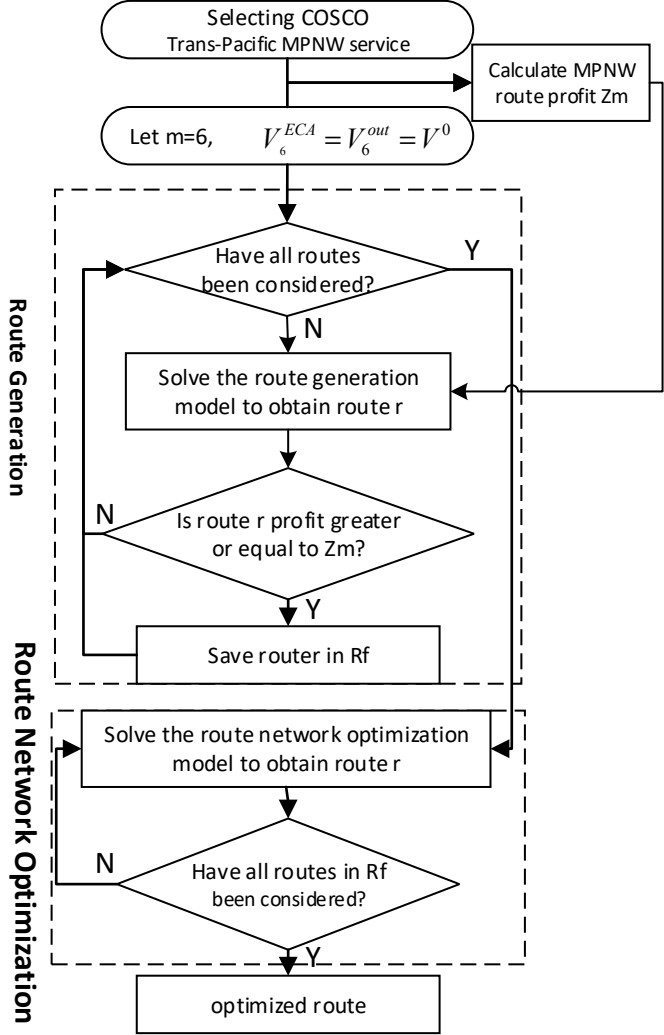

**Figure 5.** Flow chart of the heuristic algorithm based on empirical data.

### 5.2. Route Generation Model

Given the ship allocation type m and speed value $v$, the initial route is based on the existing business operations, as shown in Figure 6. We defined one vessel type, $m = 6$; $V_6^{ECA} = V_6^{ECA}out = V^0$ (i.e., defined at the economic speed of the ship) and calculated the profit (a record) for the initial route. Operating routes were based on experience and are generally the most advantageous; therefore, the Trans-Pacific-MPNW service that COSCO

is currently operating was selected as the standard of reference for the initial route. The profits of the next feasible route were then calculated according to the set parameters, and a profit comparison with the initial route was performed. If the latter was greater than the initial route profit, it was included in the set of candidate routes. This was repeated until all the routes were calculated. As shown in Figure 7, the ports marked in yellow are those with ECAs.

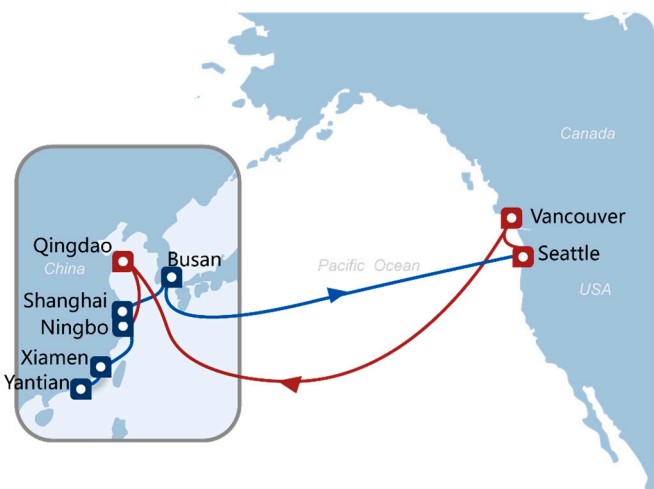

**Figure 6.** COSCOTrans-Pacific-MPNW service.

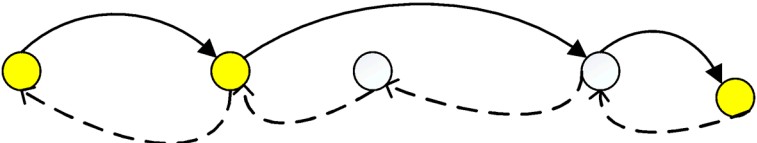

**Figure 7.** Diagram of the algorithm.

The feasible route, as shown in Figure 7, was selected to calculate the voyage.

There are four ports with ECAs, and the formula for calculating the range and total range within the ECAs is:

$$D^{ECA} = D_1^{ECA} + D_2^{ECA} + D_2^{ECA} + D_6^{ECA} + D_6^{ECA} + D_7^{ECA} + D_7^{ECA} + D_6^{ECA} + D_6^{ECA} + D_4^{ECA} + D_4^{ECA} + D_2^{ECA} + D_2^{ECA} + D_1^{ECA} \quad (31)$$

$$D = D_{12} + D_{26} + D_{67} + D_{76} + D_{64} + D_{42} + D_{21} \quad (32)$$

The total navigation time was calculated as:

$$D = D_{12} + D_{26} + D_{67} + D_{76} + D_{64} + D_{42} + D_{21} \quad (33)$$

The total time in port includes the loading and unloading time, and berthing and unberthing time, which was calculated as

$$t^r_{\text{mooring}} = \frac{2y^r_{12}}{l} + 2t_{pil} + \frac{2y^r_{26}}{l} + 2t_{pil} + \frac{2y^r_{67}}{l} + 2t_{pil} + \frac{2y^r_{76}}{l} + 2t_{pil} + \frac{2y^r_{64}}{l} + 2t_{pil} + \frac{2y^r_{42}}{l} + 2t_{pil} + \frac{2y^r_{21}}{l} + 2t_{pil} \quad (34)$$

$$T^r = t^r_{\text{sailing}} + t^r_{\text{mooring}} \quad (35)$$

Weekly ship allocations, calculated as: $n_6^r = T^r / (24 \times 7) = T^r / 168$.

Revenue of the fleet excluding stevedoring charges: $f(r)^r$, then

$$f(r)^r = \sum_{i \in N} \sum_{j \in N} \left( r_{ij} - C^l - C^u \right) \cdot y^r_{ij} \quad (36)$$

Fleet costs include the operating costs, port charges, and fuel consumption. Let the weekly operating costs of the fleet be $C_{op}^r$, the average weekly port charges be $C_p^r$ (including mooring and unmooring fees, port dues, and other fixed charges), and the average weekly fuel-consumption costs be $C_v^r$. Accordingly, the expressions for $C_{op}^r$, $C_p^r$, and $C_v^r$ are as follows.

$$C_{op}^r = 7 \cdot C_6 \cdot n_6^r \tag{37}$$

$$C_p^r = G_1^6 + G_2^6 + G_6^6 + G_7^6 + G_6^6 + G_4^6 + G_2^6 \tag{38}$$

$$F^r = F_6 \frac{D - D^{ECA}}{168v_6^0} + F_6 \frac{D^{ECA}}{168v_6^0} + A_6 \cdot n_6^r \tag{39}$$

$$C_v^r = P_{IFO} \cdot F_6 \cdot \frac{D - D^{ECA}}{168v_6^0} + P_{VLSFO} \cdot F_6 \cdot \frac{D^{ECA}}{168v_6^0} + A_6 \cdot n_6^r \cdot P_{VLSFO} \tag{40}$$

Cost of carbon emissions

$$C_{co_2}^r = \lambda \cdot F^r \cdot \delta \tag{41}$$

Single-route profit

$$f(tp)^r = f(r)^r - C_{op}^r - C_p^r - C_v^r - C_{co_2}^r \tag{42}$$

If the route profit is greater than or equal to the initial route profit, that is, if $f(tp)^r \geq f(tp)^0$, the route was added to the library, that is, $r$ was saved in $R^f$.

### 5.3. Generation of Optimal Route Networks

Optimization was performed for each route in the set of candidate routes $R_f$.

$$maxZ_{TP} = \sum_{r \in R} x^r \left\{ f(r)^r - C_{op}^r - C_p^r - C_v^r - C_{co_2}^r \right\} \tag{43}$$

Constraints: $r \in R_f$; other constraints remain unchanged.

### 5.4. Coding of the Route Generation Model

The coding for port selection and the sequence of port calls in the route generation model is a difficult part of the algorithm and is described in detail here.

On each route, a port call occurs once, forming a simple closed loop. In actual navigation, there are more complex situations than a single closed loop; for example, in a round-trip voyage, the two farthest ports are the end ports, with one port call each, while the intermediate ports can have at most two port calls. In the case of two calls, an additional virtual port is required, which was the virtual port set up in this study.

Each port has two port calls. Taking seven ports as an example, the chromosome length was set to 14, and the last seven genes represented the virtual ports. The transport route is represented by both the chromosome and gene position, with the first gene of the chromosome corresponding to gene position 1, the seventh gene to 7, etc. During decoding, the first gene position corresponds to the starting port and the next port of call is represented by the gene corresponding to the number of positions of the previous port of call. Decoding is completed when the next port-of-call number is detected as the corresponding virtual port number or the gene number corresponding to the first gene position and the virtual port number of that number, and the virtual port number is replaced with the original number. In other words, when the port of call of the ship on the round-trip route represented by this chromosome is obtained. Figure 8 shows a general circular route, and the route path is decoded as 2–5–7–2. The path does not show port calls occurring twice, and the termination of replication triggered by the decoding of this chromosome occurs when gene position 8 corresponds to chromosome 2, as shown in Figure 8.

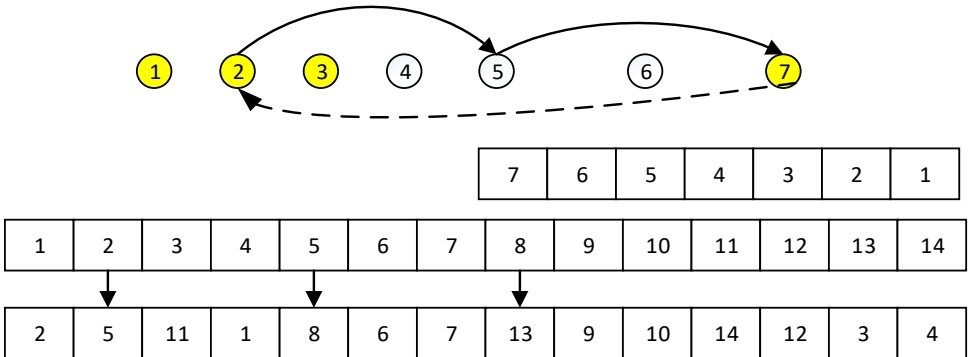

**Figure 8.** None of the port calls occurs twice.

Figure 9 shows the chromosome encoding for a situation in which a port call on the route occurs twice; the route path represented by this chromosome is 2–4–5–7–5–1–2, and the termination of the replication triggered by the decoding of this chromosome occurs when gene position 14 corresponds to chromosome 13.

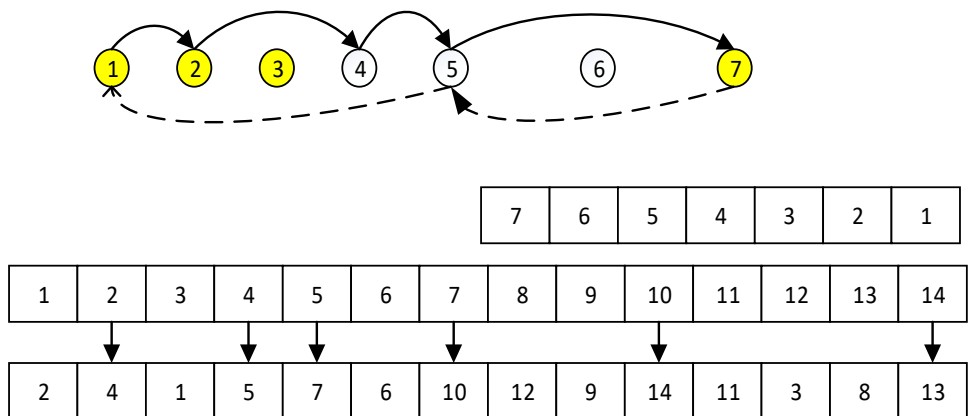

**Figure 9.** Coding of port calls that occur twice.

The design and reality of the crossover operator are closely related to the specific problem and should be considered together with the coding design because the crossover operator should be followed by a guarantee that the new chromosome can be decoded as a feasible solution. Otherwise, the crossover operator is equivalent to a simple mutation operator, which affects the learning function of the genetic algorithm for the problem and significantly reduces the convergence rate of the algorithm.

The encoding designed for the route selection problem in this study is a special ordinal encoding, which cannot be simply applied to the classical genetic-algorithm crossover operator. The following crossover rules were designed for the encoding in this study, to ensure that the generations following the crossover can be decoded as feasible solutions.

Figure 10 shows the calculation rule of the crossover operator, where useful gene positions cross other randomly assigned gene positions. In route 2–5–7–2, represented by parent 1 (Figure 8), the valid genes for this chromosome in the decoding process correspond to gene positions 2, 5, and 8 such that A = {2, 5, 8}. The route represented by parent 2 (Figure 9) is 2–4–5–7–5–1–2, and the valid genes in the decoding process for this chromosome correspond to gene positions 2, 4, 5, 7, 10, and 14 such that B = {2, 4, 5, 7, 10, 14}. With c = A∪B, the crossover is performed for the genes corresponding to all gene positions of C. The generations randomly select one of the two genes of the parent-gene positions, ensuring that the resulting generation is a feasible solution. The other gene positions that did not undergo crossover were randomly assigned.

| Position | 1 | 2 | 3 | 4 | 5 | 6 | 7 | 8 | 9 | 10 | 11 | 12 | 13 | 14 |
|---|---|---|---|---|---|---|---|---|---|---|---|---|---|---|
| Parent1 | 2 | 5 | 11 | 1 | 8 | 6 | 7 | 13 | 9 | 10 | 14 | 12 | 3 | 4 |
| Parent2 | 2 | 4 | 1 | 5 | 7 | 6 | 10 | 12 | 9 | 14 | 11 | 3 | 8 | 13 |
| Generation1 | 2 | 5 | | 1 | 7 | | 10 | 13 | | 10 | | | | 4 |
| Generation2 | 2 | 4 | | 5 | 8 | | 7 | 12 | | 14 | | | | 13 |

**Figure 10.** Crossover operator encoding.

## 6. Case Study

### 6.1. Data Collection

Hong Kong Dec 15 prices in USD are as follows: $P_{IFO} = 426.5$ USD/TON, $P_{VLSFO} = 628$ USD/TON; Shanghai Port loading and unloading efficiency, cost, and arrival and departure time are $l = 150$ TEU/h, $C^l = C^u = 65$ USD/TEU, $t_{pil} = 2$ h; reference ESMA Final Report (2022, ESMA70-445-38): $\partial = 63$ USD/TON; accord to Lowe's Marine Database, data of each ship type are shown in Table 1; accord to Ship News network Distance between ports shown in Table 2.

**Table 1.** Data of each ship type.

| Name of Vessel | m | $B_m$ (TEU) | $F_m$ (T/day) | $A_m$ (T/day) | $v_m^0$ (Kn/h) | $v_m^{min}$ (Kn/h) | $v_m^{max}$ (Kn/h) | $c_m$ (UDS/Day) | $G_m^0$ (UDS/Call) |
|---|---|---|---|---|---|---|---|---|---|
| XIN ZHANG ZHOU | 1 | 4253 | 139.5 | 6.33 | 18.2 | 11.34 | 25.15 | 9000 | 3001 |
| XIN WENZHOU | 2 | 4738 | 82 | 4.3 | 18 | 11.04 | 24.7 | 10,026 | 3344 |
| XIN YAN TIAN | 3 | 5668 | 202 | 7.81 | 17.7 | 12.05 | 26.7 | 11,994 | 4000 |
| COSCO THAILAND | 4 | 8501 | 250 | 10.47 | 18.6 | 12 | 26.6 | 17,989 | 6000 |
| XIN SHANGHAI | 5 | 9572 | 248.2 | 10.43 | 17.2 | 11.22 | 26.73 | 20,255 | 6204 |
| COSCO ASIA | 6 | 10,036 | 250 | 12.75 | 16.8 | 11.04 | 25.8 | 21,238 | 6505 |
| COSCO FAITH | 7 | 13,114 | 274.9 | 13.2 | 16.7 | 11 | 26.2 | 27,751 | 8500 |
| CSCLJUPITER | 8 | 14,074 | 262 | 14.51 | 16.1 | 11.18 | 26.62 | 29,783 | 9122 |
| CSCLPACIFIC OCEAN | 9 | 18,982 | 195.5 | 13.768 | 18 | 10 | 24.6 | 40,169 | 13,000 |
| COSCO SHIPPING VIRGO | 10 | 20,119 | 168 | 10.263 | 19 | 8.4615 | 22.5 | 42,575 | 13,040 |

Data source: Lowe's Marine Database.

**Table 2.** Distance between ports (nm).

| DKABG | CNQIN | CNSHA | CNNBO | CNXIA | CNYTN | HKHKG | JPTOK | JPOSK | SKBUS | USTAC | USVAN | USSEA |
|---|---|---|---|---|---|---|---|---|---|---|---|---|
| CNQIN | 0 | 448 | 540 | 920 | 1376 | 1362 | 1135 | 824 | 516 | 5146 | 5177 | 5127 |
| CNSHA | 448 | 0 | 243 | 623 | 1079 | 1065 | 1051 | 792 | 494 | 5124 | 5154 | 5105 |
| CNNBO | 540 | 243 | 0 | 544 | 1000 | 986 | 1062 | 832 | 542 | 5173 | 5203 | 5153 |
| CNXIA | 920 | 623 | 544 | 0 | 901 | 886 | 1374 | 1153 | 907 | 5538 | 5568 | 5518 |
| CNYTN | 1376 | 1079 | 1000 | 901 | 0 | 24 | 1752 | 1533 | 1330 | 5948 | 5974 | 5928 |
| HKHKG | 1362 | 1065 | 986 | 886 | 24 | 0 | 1738 | 1518 | 1316 | 5933 | 5959 | 5914 |
| JPTOK | 1135 | 1051 | 1062 | 1374 | 1752 | 1738 | 0 | 374 | 681 | 4316 | 4339 | 4297 |
| JPOSK | 824 | 792 | 832 | 1153 | 1533 | 1518 | 374 | 0 | 370 | 4592 | 4615 | 4573 |
| SKBUS | 516 | 494 | 542 | 907 | 1330 | 1316 | 681 | 370 | 0 | 4646 | 4676 | 4626 |
| USTAC | 5146 | 5124 | 5173 | 5538 | 5948 | 5933 | 4316 | 4592 | 4646 | 0 | 428 | 21 |
| USVAN | 5177 | 5154 | 5203 | 5568 | 5974 | 5959 | 4339 | 4615 | 4676 | 428 | 0 | 404 |
| USSEA | 5127 | 5105 | 5153 | 5518 | 5928 | 5914 | 4297 | 4573 | 4626 | 21 | 404 | 0 |

Data source: Ship News network.

The quantities used for the CA navigation distances of the routes were obtained from sea rates, and the shortest path from each port to the boundary of the ECAs was

estimated. The total number of miles navigated within the ECAs was calculated by totaling all segments, as shown in Table 3. According to China International Shipping Network, inter-port rate as shown in Table 4.

**Table 3.** ECAs voyage between ports (nm).

| DKABG | CNQIN | CNSHA | CNNBO | CNXIA | CNYTN | HKHKG | JPTOK | JPOSK | SKBUS | USTAC | USVAN | USSEA |
|-------|-------|-------|-------|-------|-------|-------|-------|-------|-------|-------|-------|-------|
| CNQIN | 0.0 | 98.6 | 87.5 | 69.3 | 88.9 | 74.5 | 40.1 | 40.1 | 40.1 | 385.6 | 305.0 | 374.4 |
| CNSHA | 98.6 | 0.0 | 105.9 | 87.7 | 107.3 | 92.9 | 58.5 | 58.5 | 58.5 | 404.0 | 323.4 | 392.8 |
| CNNBO | 87.5 | 105.9 | 0.0 | 76.6 | 96.2 | 81.8 | 47.4 | 47.4 | 47.4 | 392.9 | 312.3 | 381.7 |
| CNXIA | 69.3 | 87.7 | 76.6 | 0.0 | 78.0 | 63.6 | 29.2 | 29.2 | 29.2 | 374.7 | 294.1 | 363.5 |
| CNYTN | 88.9 | 107.3 | 96.2 | 78.0 | 0.0 | 83.2 | 48.8 | 48.8 | 48.8 | 394.3 | 313.7 | 383.1 |
| HKHKG | 74.5 | 92.9 | 81.8 | 63.6 | 83.2 | 0.0 | 34.4 | 34.4 | 34.4 | 379.9 | 299.3 | 368.7 |
| JPTOK | 40.1 | 58.5 | 47.4 | 29.2 | 48.8 | 34.4 | 0.0 | 0.0 | 0.0 | 345.5 | 264.9 | 334.3 |
| JPOSK | 40.1 | 58.5 | 47.4 | 29.2 | 48.8 | 34.4 | 0.0 | 0.0 | 0.0 | 345.5 | 264.9 | 334.3 |
| SKBUS | 40.1 | 58.5 | 47.4 | 29.2 | 48.8 | 34.4 | 0.0 | 0.0 | 0.0 | 345.5 | 264.9 | 334.3 |
| USTAC | 385.6 | 404.0 | 392.9 | 374.7 | 394.3 | 379.9 | 345.5 | 345.5 | 345.5 | 0.0 | 418.4 | 6.0 |
| USVAN | 305.0 | 323.4 | 312.3 | 294.1 | 313.7 | 299.3 | 264.9 | 264.9 | 264.9 | 418.4 | 0.0 | 412.4 |
| USSEA | 374.4 | 392.8 | 381.7 | 363.5 | 383.1 | 368.7 | 334.3 | 334.3 | 334.3 | 6.0 | 412.4 | 0.0 |

**Table 4.** Inter-port rate (USD/TEU).

| DKABG | CNQIN | CNSHA | CNNBO | CNXIA | CNYTN | HKHKG | JPTOK | JPOSK | SKBUS | USTAC | USVAN | USSEA |
|-------|-------|-------|-------|-------|-------|-------|-------|-------|-------|-------|-------|-------|
| CNQIN | 0 | 122 | 147 | 250 | 374 | 371 | 309 | 224 | 140 | 1400 | 1400 | 1350 |
| CNSHA | 122 | 0 | 66 | 169 | 294 | 290 | 287 | 216 | 135 | 1400 | 1400 | 1350 |
| CNNBO | 147 | 66 | 0 | 148 | 272 | 268 | 287 | 225 | 147 | 1400 | 1400 | 1350 |
| CNXIA | 250 | 169 | 148 | 0 | 245 | 241 | 385 | 323 | 254 | 1550 | 1550 | 1500 |
| CNYTN | 374 | 294 | 272 | 245 | 0 | 7 | 200 | 250 | 347 | 1550 | 1550 | 1500 |
| HKHKG | 371 | 290 | 268 | 241 | 7 | 0 | 100 | 100 | 344 | 1550 | 1550 | 1500 |
| JPTOK | 309 | 286 | 289 | 374 | 477 | 473 | 0 | 100 | 178 | 1128 | 1129 | 1090 |
| JPOSK | 224 | 215 | 226 | 314 | 417 | 413 | 102 | 0 | 101 | 1200 | 1200 | 1160 |
| SKBUS | 140 | 134 | 147 | 247 | 362 | 358 | 185 | 101 | 0 | 1214 | 1216 | 1173 |
| USTAC | 1400 | 1394 | 1407 | 1507 | 1618 | 1614 | 1174 | 1249 | 1264 | 0 | 116 | 6 |
| USVAN | 1408 | 1402 | 1416 | 1515 | 1625 | 1621 | 1180 | 1256 | 1272 | 116 | 0 | 110 |
| USSEA | 1395 | 1389 | 1402 | 1501 | 1613 | 1609 | 1169 | 1244 | 1259 | 6 | 110 | 0 |

The OD demand data between ports refers to a weekly demand of approximately 200–1000 TEU from an Asian port to an American port (round trip), and the OD demand data were randomly generated, as shown in Table 5.

**Table 5.** Inter-port OD demand (TEU/week).

| DKABG | CNQIN | CNSHA | CNNBO | CNXIA | CNYTN | HKHKG | JPTOK | JPOSK | SKBUS | USTAC | USVAN | USSEA |
|-------|-------|-------|-------|-------|-------|-------|-------|-------|-------|-------|-------|-------|
| CNQIN | 0 | 0 | 0 | 0 | 0 | 0 | 446 | 724 | 849 | 320 | 965 | 423 |
| CNSHA | 0 | 0 | 0 | 0 | 0 | 0 | 235 | 304 | 383 | 409 | 794 | 851 |
| CNNBO | 0 | 0 | 0 | 0 | 0 | 0 | 663 | 305 | 459 | 840 | 379 | 954 |
| CNXIA | 0 | 0 | 0 | 0 | 0 | 0 | 235 | 686 | 322 | 219 | 869 | 380 |
| CNYTN | 0 | 0 | 0 | 0 | 0 | 0 | 632 | 325 | 623 | 810 | 383 | 986 |
| HKHKG | 0 | 0 | 0 | 0 | 0 | 0 | 417 | 524 | 816 | 954 | 705 | 540 |
| JPTOK | 874 | 493 | 818 | 745 | 500 | 762 | 0 | 0 | 0 | 871 | 479 | 611 |
| JPOSK | 615 | 665 | 562 | 293 | 862 | 342 | 0 | 0 | 0 | 296 | 389 | 666 |
| SKBUS | 535 | 977 | 211 | 787 | 457 | 647 | 0 | 0 | 0 | 721 | 559 | 929 |
| USTAC | 556 | 473 | 651 | 691 | 427 | 958 | 505 | 917 | 696 | 0 | 0 | 0 |
| USVAN | 936 | 239 | 972 | 432 | 843 | 818 | 470 | 716 | 997 | 0 | 0 | 0 |
| USSEA | 584 | 997 | 726 | 235 | 826 | 632 | 587 | 584 | 916 | 0 | 0 | 0 |

### 6.2. Analysis of Results

According to the above algorithm design, the population size N = 100, crossover probability Pc = 0.8, mutation probability Pm = 0.02, and maximum number of iterations G = 200 were obtained. To verify the validity of the model and algorithm, four different calculation methods were used. The first method is based on the optimization of existing routes, that is, the optimization of the initial route in the algorithm design, the Trans-Pacific-MPNW service (denoted as MPNW-LRO). The second method is based on the network design based on the design speed (denoted as DES-LSND), that is, optimization is performed using the design speed of each type of vessel. The third method is a liner-shipping network design based on equal speed inside and outside the ECAs (denoted as ECA-LSND), that is, $V^{ECA} = V^{out}$ for optimization. Finally, the fourth method (denoted as LSND) is the liner-shipping network optimization in this study. The ports are coded for statistical purposes, as shown in Table 6; the calculation results are presented in Table 7; and the optimized transport between ports is shown in Table 8.

**Table 6.** Port number.

| CNQIN: | CNSHA: | CNNBO: | CNXIA: | CNYTN: | HKHKG: | JPTOK: | JPOSK: | SKBUS: | USTAC: | USVAN: | USEA: |
|---|---|---|---|---|---|---|---|---|---|---|---|
| 1 | 2 | 3 | 4 | 5 | 6 | 7 | 8 | 9 | 10 | 11 | 12 |

**Table 7.** Calculation results.

| Method | LSND | DES-LSND | ECA-LSND |
|---|---|---|---|
| Port of call and order | 5-6-4-3-2-9-8-7-11-12-10-7-8-9-1-2-4-6-5 | 5-6-4-3-2-9-8-7-11-12-10-7-8-9-1-2-4-6-5 | 5-6-4-3-2-9-8-7-11-12-10-7-8-9-1-2-4-6-5 |
| Profit | 3,957,362.95 | 2,438,426.32 | 3,870,687.40 |
| Total cost | 1,686,897.72 | 1,851,594.22 | 1,722,952.43 |
| Fuel cost | 691414.00 | 847,338.83 | 689,188.20 |
| Carbon emission | 7839.08 | 10,382.49 | 7812.62 |
| In the ECA speed | 13.535 | 18.00 | 14.03 |
| ECA outside speed | 14.627 | 18.00 | 14.03 |
| Ship type | 8 | 8 | 8 |
| Ship number | 7 | 7 | 7 |

**Table 8.** Optimized inter-port traffic volume.

| DKABG | CNQIN | CNSHA | CNNBO | CNXIA | CNYTN | HKHKG | JPTOK | JPOSK | SKBUS | USTAC | USVAN | USSEA |
|---|---|---|---|---|---|---|---|---|---|---|---|---|
| CNQIN | 0 | 0 | 0 | 0 | 0 | 0 | 0 | 0 | 0 | 0 | 0 | 0 |
| CNSHA | 0 | 0 | 0 | 0 | 0 | 0 | 923 | 451 | 867 | 225 | 597 | 483 |
| CNNBO | 0 | 0 | 0 | 0 | 0 | 0 | 100 | 137 | 136 | 293 | 161 | 110 |
| CNXIA | 0 | 0 | 0 | 0 | 0 | 0 | 177 | 99 | 235 | 795 | 577 | 476 |
| CNYTN | 0 | 0 | 0 | 0 | 0 | 0 | 739 | 768 | 394 | 379 | 314 | 808 |
| HKHKG | 0 | 0 | 0 | 0 | 0 | 0 | 369 | 935 | 902 | 331 | 799 | 205 |
| JPTOK | 716 | 556 | 0 | 208 | 596 | 453 | 0 | 0 | 0 | 784 | 773 | 844 |
| JPOSK | 167 | 250 | 0 | 134 | 90 | 644 | 0 | 0 | 0 | 758 | 859 | 768 |
| SKBUS | 567 | 356 | 0 | 197 | 575 | 248 | 0 | 0 | 0 | 446 | 918 | 438 |
| USTAC | 652 | 312 | 0 | 782 | 584 | 804 | 457 | 380 | 354 | 0 | 0 | 0 |
| USVAN | 172 | 272 | 0 | 656 | 760 | 238 | 637 | 162 | 442 | 0 | 0 | 0 |
| USSEA | 494 | 575 | 0 | 538 | 481 | 350 | 791 | 279 | 825 | 0 | 0 | 0 |

The arithmetic validation presented above shows the following:

(1) The thesis affirms that regarding the liner-shipping network optimization model that considers the environmental costs and the design of the heuristic algorithm based on empirical data, the validity of the model and algorithm can be proven through empirical calculations, which provide an effective solution for the liner network design

of shipping companies. The overall better route design (port of call and sequence of port calls), route allocation (vessel type, fleet size, and speed inside and outside the ECAs region), and cargo transportation plan (cargo transport between ports) were obtained through calculations.

(2) A comparison between DES-LSND and LSND shows that the LSND method is 5.02% more profitable and 10.79% less carbon-intensive than the DES-LSND method. With the carbon tax on international shipping and the successive introduction of increasingly stringent policies related to environmental protection in maritime transport, slow steaming has become a major trend, and represents a more effective method for shipping companies to adapt to environmental-protection requirements. It also shows that the LSND method proposed in this study considers the problem from the perspective of global optimization and can lead to a significant increase in the profitability of the route network compared to the route network obtained at design speed (DES-LSND). This is because the DES-LSND method uses the design-speed value; hence, the cost of each segment is fixed. Therefore, when market conditions change, the optimization results can only adjust the route network, and it is more difficult to obtain a globally optimal solution.

(3) The comparison between ECA-LSND and LSND shows that slower steaming within the LSND emission control area (two speeds) is 2.24% more profitable and 0.34% more carbon-intensive than the single-speed strategy (one speed) of ECA-LSND. This further shows that when the market conditions (introduction of the ECA policy) change, the LSND approach takes a more holistic view of the problem and results in a significant increase in the profitability of the route network compared with the route network obtained with a single-speed strategy (ECA-LSND). This is because the ECA-LSND method uses a single-speed strategy inside and outside the ECAs, thus making it harder to obtain a globally optimal solution because the ship does not adjust its speed in response to the high price of the fuel that must be used in the ECAs, leading to an increase in fuel costs. These ECAs increase carbon emissions, which have a negative impact on marine environments. The ECA policy is mainly designed to reduce sulfur emissions, but it also leads to an increase in carbon emissions. This is also the reason why ECAs emission control areas are called SECA in many places, as well as the defect of the ECA policy.

(4) Under the emission-control-area policy, it is the best sailing strategy for shipping companies to adopt different sailing speeds outside the ECAs and low sailing speeds within the ECAs. Ships entering ECAs using high-priced low-sulfur marine gas oil can minimize total costs and increase profits by slowing down [16]. As increasing attention is being paid to marine environmental protection, speed optimization inside and outside the ECAs will become increasingly important. Shipping companies must pay attention to the influence of ECAs and optimize the speed inside and outside ECAs when making sailing plans.

### 6.3. Sensitivity Analysis

An analysis of the impact of fluctuations in port OD demand, tariffs, and ECAs on the route network and its profitability and a comparison of the adjustment scheme and changes in total profitability of the route network for the different methods mentioned above were performed.

### 6.3.1. Sensitivity Analysis of Emission-Control-Area Numbers

To further investigate whether the implementation of the ECA policy affects the design of shipping-companies' transportation networks, this paper added "port ECAs" to the existing ECAs. Case 1 allows all ports to implement emissions controls, and in Case 2, all port ECAs are withdrawn. The results are shown in the Table 9.

**Table 9.** Results of sensitivity analysis of emission control areas numbers.

| Method | Case 1 | Case 2 | LSND |
|---|---|---|---|
| Port of call and order | 5–6–4–3–2–9–8–7–11–12–10–7–8–9–1–2–4–6–5 | 5–6–4–3–2–9–8–7–11–12–10–7–8–9–1–2–4–6–5 | 5–6–4–3–2–9–8–7–11–12–10–7–8–9–1–2–4–6–5 |
| Profit | 21,280,885.56 | 2,733,597.50 | 3,957,362.95 |
| Total cost | 6,869,693.11 | 2,663,319.99 | 1,686,897.72 |
| Fuel cost | 1,360,256.79 | 862,967.65 | 691,414.00 |
| Carbon emission | 8416.53 | 13,697.90 | 7839.08 |
| In the ECA speed | 12.133 | 14.39 | 13.535 |
| ECA outside speed | 14.872 | | 14.627 |
| Ship type | 8 | 8 | 8 |
| Ship number | 7 | 7 | 7 |

When Scenario 1 was compared with LSND, without a change in the choice and order of ports of call, profit decreased by 1.65%, and carbon emissions increased by 7.78%. When scenario 2 was compared with LSND, without a change in the choice and order of ports of call, profit increased by 2.43%, and carbon emissions increased by 2.58%. The results show that the implementation of the ECA policy has had little impact on the profits of shipping companies and transport networks. Although the establishment of EACs will, to some extent, increase the cost burden on shipowners, the impact of ECAs on costs is relatively small compared with the total. Thus, the ECAs policy is shown to have little impact on the attractiveness and competitiveness of ports [30]. In addition, it is again proven that ECAs can lead to increased carbon emissions.

The policy regime is faced with the dilemma of port development and environmental protection, and the results of this study show that the ECAs policy has less impact on the attractiveness and competitiveness of ports, while the supply and demand of port cargo (i.e., port throughput) is the key factor in attracting ships.

6.3.2. Analysis of OD Demand between Ports and Tariff Sensitivity

To further investigate whether OD demand between ports and freight rate fluctuations affect the design of shipping-companies' transportation networks, OD demand and freight rate fluctuations were reduced by 20%, and the results are shown in Table 10.

**Table 10.** Results of OD demand between ports and freight rate sensitivity analysis.

| Method | Demand Is Reduced by 20% | Freight Rates Were Reduced by 20% | Invariant |
|---|---|---|---|
| Port of call and order | 5–6–4–3–2–9–8–7–11–12–10–7–8–9–1–2–4–6–5 | 5–4–2–9–8–7–11–12–7–8–9–1–2–4–5 | 5–6–4–3–2–9–8–7–11–12–10–7–8–9–1–2–4–6–5 |
| Profit | 2,854,323.595 | 2,978,286.01 | 3,957,362.95 |
| Fuel cost | 697,085.77 | 649,626.763 | 691,414.00 |
| Carbon emission | 7906.49 | 7279.2 | 7839.08 |
| In the ECA speed | 13.68 | 11.86 | 13.535 |
| ECA outside speed | 14.42 | 13.01 | 14.627 |
| Ship type | 8 | 8 | 8 |
| Ship number | 7 | 7 | 7 |

The results show that profit decreased by 27.87% with a 20% reduction in demand and by 24.74% with a 20% reduction in freight rates. In both cases, there is a decrease in port calls, and the decrease in port calls is more prominent in the case of a 20% decrease in freight rates. Therefore, increasing port cargo volume is crucial in increasing port attractiveness and competitiveness; that is, port hinterland development and expansion lead to an increase in port cargo volume, which naturally increases port attractiveness [35].

### 6.3.3. Sensitivity Analysis of Different Carbon Prices

To further determine whether the fluctuation in carbon prices (assuming that the carbon tax is equal to the carbon price) affects the transportation network design of shipping companies, the carbon prices were reduced by 20% and increased by 20%. The results are presented in Table 11.

**Table 11.** Results of different carbon prices sensitivity analysis.

| Method | Carbon Prices 50.4 | Carbon Prices 63 | Carbon Prices 75.6 |
|---|---|---|---|
| Port of call and order | 5–6–4–3–2–9–8–7–11–12–7–8–9–1–2–4–5 | 5–6–4–3–2–9–8–7–11–12–10–7–8–9–1–2–4–6–5 | 5–4–3–2–9–8–7–11–12–7–8–9–1–2–4–5 |
| Profit | 4,021,694.133 | 3,957,362.95 | 3,866,976.2 |
| Fuel cost | 804,827.37 | 691,414 | 556,354.72 |
| Carbon emission | 8800.1011 | 7839.08 | 6634.17473 |
| In the ECA speed | 14.14 | 13.535 | 12.78 |
| ECA outside speed | 14.89 | 14.627 | 14.097 |
| Ship type | 8 | 8 | 8 |
| Ship number | 6 | 7 | 7 |

Carbon prices directly affect profits and emissions. High carbon prices reduce profits and carbon emissions, whereas low carbon prices increase carbon emissions. If carbon trading is carried out, it can motivate shipping companies to reduce carbon emissions under the condition of high carbon prices and increase revenue and profits through carbon trading [14].

### 6.3.4. Sensitivity Analysis of Different Fuel Prices

To further determine whether the fluctuation in fuel cost affects the transportation network design of shipping companies, the price fluctuation in fuel cost was reduced by 10% and 20%, and increased by 10% and 20%, respectively. The results are shown in Table 12.

**Table 12.** Results of different-fuel-price sensitivity analysis.

| Method | Lower Fuel Prices 20% | Lower Fuel Prices 10% | LSND | Increase in Fuel Prices 10% | Increase in Fuel Prices 10% |
|---|---|---|---|---|---|
| Port of call and order | 5–6–4–3–2–9–8–7–11-12–10–7–8–9–1–2–4–6–5 | 5–6–4–3–2–9–8–7–11-12–10–7–8–9–1–2–4–6–5 | 5–6–4–3–2–9–8–7–11-12–10–7–8–9–1–2–4–6–5 | 5–6–4–3–2–9–8–7–11-12–10–7–8–9–1–2–4–6–5 | 5–4–3–2–9–8–7–11–12–10–7–8–9–1–2–4–5 |
| Profit | 4,210,848.61 | 4,002,766.25 | 3,957,362.95 | 3,652,554.54 | 3,010,744.785 |
| Fuel cost | 606,727.61 | 626,287.67 | 691,414 | 722,529.43 | 741,462.76 |
| Carbon emission | 8219.19 | 8069.97 | 7839.08 | 6833.99 | 6962.50 |
| In the ECA speed | 14.27 | 12.37 | 13.53 | 11.78 | 11.96 |
| ECA outside speed | 14.66 | 14.77 | 14.63 | 13.45 | 12.21 |
| Ship type | 8 | 8 | 8 | 8 | 8 |
| Ship number | 6 | 7 | 7 | 7 | 7 |

At present, most shipping companies use changes in the ship speed to control costs. Low oil prices play a role in reducing the costs of shipping companies. When oil prices drop, they choose to increase ship speed, reduce ship investment, or increase the voyage number. At the same time, it was found that low oil prices lead to higher carbon emissions. When oil prices drop, shipping companies choose to increase their shipping speed, which can offset the cost of increased carbon emissions by increasing voyage profits or reducing ship investment. Therefore, a low oil price has a negative impact on energy conservation and emission reduction [36].

*6.4. Management Implications*

1. With the introduction of the global maritime carbon tax and increasingly strict maritime-transport environmental-protection policies, low-speed navigation has become the trend of the times [14], and it is also a relatively effective way for shipping companies to deal with environmental protection.

2. As increasing attention is being paid to marine environmental protection, speed optimization will become increasingly important. Shipping companies must pay attention to the influence of ECAs and optimize the speed inside and outside ECAs when making sailing plans [16]. For shipping companies, low-speed sailing in ECAs is the best strategy.

3. The ECA policy system faces the dilemmas of port development and environmental protection. Studies show that the implementation of the ECA policy has little influence on the design of the ship transport network; therefore, ECAs have little influence on port competitiveness [30], and the supply and demand of port goods, namely, port throughput, is the key factor in attracting ships [35].

4. The main function of ECAs is to reduce sulfur emissions, but this will lead to an increase in carbon emissions. These ECAs should be combined with carbon-emissions control policies. Only by implementing dual policies can the marine environment be better protected. At the same time, the development of the shipping market should also be considered when formulating policies to develop reasonable policies.

## 7. Conclusions

This study built a liner-shipping network optimization model that considers environmental costs and designed a heuristic algorithm for empirical data to solve it. Additionally, the validity of the model and algorithm was verified by assessing the actual risks, and the following conclusions were reached.

- This study proposes a liner-shipping network optimization model considering environmental costs, designs a heuristic algorithm based on empirical data, and proves the effectiveness of the model and algorithm through empirical calculations, providing an effective solution for the liner network design of shipping companies. Liner companies can use the model and algorithm proposed in this study to readjust and optimize the linear network when market conditions change, thus simultaneously preserving the market share and maximizing the profit of the liner network.

- The LSND method proposed in this study considers the problem from the perspective of global optimization and is more likely to yield globally optimized solutions than the other methods, and provides a new reference for solving liner transport network design problems.

- With the introduction of a global maritime carbon tax and increasingly strict maritime-transport environmental-protection policies, more and more attention has been paid to marine environmental protection, and speed optimization will become more and more important. Shipping companies must pay attention to the influence of ECAs and make overall optimization of ECA internal and external speed when formulating navigation plans. For shipping companies, low-speed sailing in ECA areas is the best sailing strategy [14].

- The impact of fluctuations in freight rates on shipping-company profits and transport networks is most significant, followed by the impact of OD demand between port fluctuations. The implementation of the ECA policy has had relatively little impact on shipping-company profits and transport networks [30]. When a policy regime balances port development and environmental protection, the key to increasing the attractiveness and competitiveness of ports is to increase the volume of port cargo [35].

- Low oil prices will have a negative impact on energy conservation and emission reduction [36]. Most shipping companies cut costs by sailing at lower speeds, but when oil prices drop, shipping companies choose to increase shipping speed, which

can offset the cost of increased carbon emissions by increasing voyage profits or reducing ship investment.

Future research can include weather factors, as the complex and changeable weather situation influence the weather on the ship voyage, and has the significant characteristics of randomness, and being real-time and dynamic. In the future, environmental factors can be considered to build a variable-speed transport optimization model affected using random factors.

**Author Contributions:** Conceptualization and methodology, X.L., Q.T. and X.W.; writing—original draft preparation, X.L. and Q.T; writing—review and editing, X.L.; software, X.L. and X.W.; supervision, Q.T. All authors have read and agreed to the published version of the manuscript.

**Funding:** Please add: This research was funded by the Marine Economy Development Special Fund Project of Guangdong Province—Guangdong Natural Resources Cooperation [2020], grant number 071.

**Institutional Review Board Statement:** Not applicable.

**Informed Consent Statement:** Not applicable.

**Data Availability Statement:** Not applicable.

**Conflicts of Interest:** The authors declare no conflict of interest.

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
