# Peer review of "Liner-Shipping Network Design with Emission Control Areas: A Real Case Study"

_sustainability, doi:10.3390/su15043734_

Round 1

Reviewer 1 Report

The paper is generally well written and clear. The topic is important and the treatment contains a number of interesting features including the construction of a liner shipping network design model that includes package cargo transport plan, route allocation and route design with the objective to maximize the profit by selecting the ports to be visited, the cargo flows and the number/operating speeds of vessels. A crucial element that is also considered in this work is the impact of ECAs in the existing liner network. The methodology used is well described and the results are made clear to the reader. The literature review, though, is not comprehensive, at least with regards to existing literature on the impact of ECAs on shipping operations as well as speed optimization.  

The literature on the topic is quite rich and I can refer to a number of relevant previous works that could be inserted in the analysis:

- Brynolf, S., Magnusson, M., Fridell, E., & Andersson, K. (2014). Compliance possibilities for the future ECA regulations through the use of abatement technologies or change of fuels. Transportation Research Part D: Transport and Environment, 28, 6-18.
- Chen, L., Yip, T. L., & Mou, J. (2018). Provision of Emission Control Area and the impact on shipping route choice and ship emissions. Transportation Research Part D: Transport and Environment, 58, 280-291.

Additionally, there are certain statements in the paper that are not supported by previous works. More specifically,  

1)    Line 55: ‘The price of clean fuel is bound to be much higher …’ Indicative literature that could support this statement:

Pomaska, L., & Acciaro, M. (2022). Bridging the Maritime-Hydrogen Cost-Gap: Real options analysis of policy alternatives. Transportation Research Part D: Transport and Environment, 107, 103283.

2)    Line 655: The authors are discussing the introduction of a global barbon levy, but have not made any reference to any IMO documents or academic literature. Impacts of a bunker levy on decarbonizing shipping: A tanker case study. Transportation Research Part D: Transport and Environment, 106, 103257; Christodoulou, A., Dalaklis, D., Ölcer, A., & Ballini, F. (2021). Can market-based measures stimulate investments in green technologies for the abatement of ghg emissions from shipping? A review of proposed market-based measures. Transactions on Maritime Science, 10(01), 208-215).

3)    There is a mistake in line 50 stating that the IMO is a regional organization. However, the IMO is a global organization. Please check/cite ‘Christodoulou, A., & Echebarria Fernández, J. (2021). Maritime Governance and International Maritime Organization instruments focused on sustainability in the light of United Nations’ sustainable development goals. In Sustainability in the Maritime Domain (pp. 415-461). Springer, Cham’.

I hope my comments and additions are helpful for the authors and would support them in the improvement of their paper to make it suitable for publication in this journal.

Author Response

1)Line 55:----Revised, (line 55-56)

2)Line 655:Revised,(line 334).There are two main potential carbon emissions policies available: the cap-and-trade programme and the carbon tax (dy, Pizer 2015; Yang et al. 2017; Xing, Y. et al.2019) [32,33,14]

3)Revised, (line 52)

Reviewer 2 Report

The paper needs to make improvements. 

1. I suggest the title using the long form of ECAs rather than the short form. Also, a real case study is too abstract. Please revise it. 

2. Please conduct the editing of the manuscript. 

3. I have seen some terms using CO2 which is not consistent. Please make double-check.

4. I have seen the format of citations is not alignment with the journal requirement. Please make checking it. 

5. Start from the discussion to the conclusion section, there is no literature to support. Please provide the references to support the argument. Also, the academic contribution needs to improve in the conclusion section. 

6. I have seen that the paper only uses 31 references. But I have seen the in-text reference number has 89. Please correct the mistake. 

Author Response

  1. Revised
  2. The manuscript was re-edited
  3. Has been checked.   There are two main potential carbon emissions policies available: the cap-and-trade programme and the carbon tax (dy, Pizer 2015; Yang et al. 2017;Xing, Y. et al.2019) [32,33,14]
  4. Revised
  5. Revised
  6. Revised,(line 203)

Round 2

Reviewer 1 Report

The authors have adequately addressed my comments; therefore, I believe the paper in its current form is suitable for publication in this journal.

Author Response

Thanks

Reviewer 2 Report

The paper has improved a lot. Please provide the appropriate references to support the argument in Sections 6.4 and 7, respectively.

Author Response

Revised